# Recognition of Cross-Language Acoustic Emotional Valence Using Stacked Ensemble Learning

**Kudakwashe Zvarevashe** and **Oludayo O. Olugbara** *

ICT and Society Research Group, South Africa Luban Workshop, Durban University of Technology, Durban 4001, South Africa; 21752377@dut4life.ac.za

* Correspondence: oludayoo@dut.ac.za

**Abstract:** Most of the studies on speech emotion recognition have used single-language corpora, but little research has been done in cross-language valence speech emotion recognition. Research has shown that the models developed for single-language speech recognition systems perform poorly when used in different environments. Cross-language speech recognition is a craving alternative, but it is highly challenging because the corpora used will have been recorded in different environments and under varying conditions. The differences in the quality of recording devices, elicitation techniques, languages, and accents of speakers make the recognition task even more arduous. In this paper, we propose a stacked ensemble learning algorithm to recognize valence emotion in a cross-language speech environment. The proposed ensemble algorithm was developed from random decision forest, AdaBoost, logistic regression, and gradient boosting machine and is therefore called RALOG. In addition, we propose feature scaling using random forest recursive feature elimination and a feature selection algorithm to boost the performance of RALOG. The algorithm has been evaluated against four widely used ensemble algorithms to appraise its performance. The amalgam of five benchmarked corpora has resulted in a cross-language corpus to validate the performance of RALOG trained with the selected acoustic features. The comparative analysis results have shown that RALOG gave better performance than the other ensemble learning algorithms investigated in this study.

**Keywords:** deep learning; ensemble learning; feature elimination; feature selection; speech emotion; speech recognition

## 1. Introduction

The advent of machine learning methods has brought about groundbreaking research and ubiquitous applications of different algorithms to solve challenging problems facing humans. Nowadays, we live in a world where the idea of detecting skin melanoma [1] and plant diseases [2] is made possible because of the innovation of machine learning. The field of communication network has not been spared in the pervasive adoption of intelligent systems for supporting human decision-making process. The necessity to develop intelligent robotic systems has triggered innovative research in many fields such as the healthcare, industry automation [3], human–computer interaction [4], and many other practical application areas. Embedding emotional intelligence in business units such as customer call centers is providing a better alternative of measuring customer satisfaction and evaluation of agents [5]. The development of a model that can recognize customer satisfaction requires a blend of different languages and accents. Developing systems that are emotionally intelligent requires an appropriate model to represent emotions [6]. The categorical approach and dimensional approach are two possible ways of modeling human emotions [7]. Categorical models divide emotions into several categories such as sadness, surprise, fear, and so on [7]. The dimensional approach presents speech emotions in different dimensions such as the three dimensions of arousal, dominance, and valence [8].

Arousal describes how excited or apathetic an individual gets when a particular event triggers [9]. Dominance describes submission, while polarity-based valence speaks to the positivity and negativity of human emotions [10].

This research focuses on valence emotion, which is one of the most significant scientific concepts that lies at the core of emotional experience. Emotional valence expresses the pleasantness or hedonic value of emotional states and classifies them appositely as positive or negative [11]. In fact, valence can be used to evaluate user experience in human–computer interactivity [12]. Moreover, it can be used to determine the satisfaction, contentment, and approval of clients in a customer call center environment [13]. A customer may register a grievance at a call center showing discontent or disenchantment with regard to the service offered but may end up expressing feelings of joy or contentment after receiving favorable feedback from the call center agent. A call center agent might be efficient and soft spoken and thereby fulfilling the basic functional needs of the user. However, if it creates a bad experience, the users obviously would not like it, which can create emotional acrimony or a barrier. Therefore, this research work is focused on valence speech emotion recognition (VSER) with the purpose of extracting discriminative acoustic features from speech signals to improve recognition performance [14]. The application of such a recognition system will not only enable us to comprehend the meaning of the messages being conveyed, but to also know how messages are conveyed [15]. Rong et al. [16] have suggested unequivocally that for listeners to get a deeper comprehension of messages communicated, there is the need to understand the emotional states of the speakers involved in the communication.

Some researchers have fused facial and speech features to improve the recognition of depression in an individual [17,18]. However, many of the previous studies done in VSER have been conducted using monolingual emotion corpus such as Surrey audio-visual expressed emotion (SAVEE) [19] and the Berlin database of emotional speech (EMO-DB) [20]. Models developed using monolingual corpora have performed well when tested on such architypes of corpora, but they often perform poorly on multilingual emotion corpora [21,22]. Moreover, research has shown that speech emotion models perform better when applied under the conditions that are analogous to the original environment and circumstance [23,24]. In addition, cross-language communication is beneficial in most customer call centers in African countries where multiple languages are official languages approved by the government. For example, in Zimbabwe, there are 16 official languages, and multilingual support in Shona, Ndebele, English, and Tonga are frequently used at call centers for communication. Such an environment demands the development of a versatile cross-language VSER model instead of a separate speech recognition model for each official language. Hence, there is the need to develop speech emotion models using several corpora with different languages, cultures, and recording environments. This novel approach is widely referred to as cross-language speech emotion recognition, which is another focus of this study.

Cross-language speech emotion recognition is a difficult task because of the sparseness and intrinsic dissimilarities among speech emotion corpora [25]. Most of the existing corpora consist of different languages, accents, and speakers from different culture and heritage [26]. Speech emotions are generally recorded under varying conditions and environments that can be noisy or proliferated with undesirable artifacts [24]. The quality of equipment used for recording the vocal utterances may be different, which can inhibit speech recognition performance [27]. Differential in the gender bigotry and age chauvinism of speakers can also impede speech recognition performance [28]. There are three standard types of emotion corpora that are frequently exploited for speech recognition experiments, which are acted, elicited, and natural [29]. The techniques used in creating speech emotion recognition corpora can significantly affect the output of the recognition systems [23]. These are the critical factors that can affect the performance of a speech recognition system when tested on different corpora. Consequently, there is a necessity to investigate the most discriminative speech features and effective recognition methods that can help recognize speech emotions in multilingual settings.

The RALOG ensemble learning algorithm proposed in this paper—a combination of random decision forest, AdaBoost, logistic regression, and gradient boosting machine—can effectively recognize multilingual positive and negative emotional valences. The choice of an ensemble algorithm in this study was inspired by the success story of various practical applications of ensembles such as lung cancer prediction [30], gender voice identification [31], hyperspectral image classification [32], and product image classification [33]. Therefore, this research work makes the following distinctive contributions to the scientific body of knowledge.

- The selection of highly discriminating features from an existing set of acoustic features using a random forest recursive feature elimination (RF-RFE) algorithm to seamlessly recognize positive and negative valence emotions in a cross-language environment is a unique contribution of this study. Most of the existing methods reported in the literature hardly use feature selection for cross-language speech recognition, yet it is essential to identify the features that can easily distinguish valence emotions across different languages and accents.
- The development of a RALOG-staked ensemble learning algorithm that can effectively recognize positive and negative valence emotions in human spoken utterances with high precision, recall, F1 score, and accuracy values is an important contribution of this study. Moreover, the training time of the algorithm was greatly reduced through feature scaling to improve the efficiency of the algorithm.
- The experimental comparison of various benchmarked ensemble learning algorithms to test the effectiveness of the proposed RALOG algorithm based on the selected acoustic features is an essential contribution of this research work.

The remainder of this paper is concisely summarized as follows. The related literature is discussed in Section 2. The study materials and methods are presented in Section 3. Section 4 converses the results of the study, and the paper is succinctly concluded in Section 5.

## 2. Related Literature

Developing cross-language speech emotion recognition systems has been a hot research agenda in the field of human–machine interaction over the past few years. Researchers in the speech recognition area have proposed different methods, including feature normalization [34] and decision fusion [35] to improve the performance of speech emotion recognition systems. However, many of the research studies done in speech emotion recognition have been mainly focused on single-language speech recognition systems. Relatively very few studies have been done in cross-language speech emotion recognition systems that can be useful in a language immersion environment with the goal of constructing an intelligent machine that can understand and speak multiple human languages.

The authors in [36] used a methodology based on the Geneva minimalistic acoustic parameter set (eGeMAPS), deep belief network (DBN)-based transfer learning, and leave-one-out (LOO) to improve the performance of a speech recognition system for the case of a cross-language application. They trained their model using the LOO technique with interactive emotional dyadic motion capture (IEMOCAP), EMO-DB, Italian emotional speech database (EMOVO), and SAVEE. They used the Friedrich Alexander Universität artificial intelligence bot (FAU-Aibo) emotion corpus to test their model that achieved an overall accuracy of 80% [22]. Latif et al. [25] used the EMOVO, Urdu speech dataset, SAVEE, and EMO-DB corpora of utterances spoken in Italian, Urdu, English, and German languages, respectively to develop a cross-language speech emotion recognition model. They used eGeMAPS with support vector machine (SVM) to classify the resulting emotions into positive and negative valences according to the emotion mapping shown in Table 1. Their cross-language valence speech classification model achieved an overall accuracy of 70.98%.

**Table 1.** Comparative analysis of our method with related methods.

| Reference | Corpora | Languages | Emotion Mapping | Recognition Method | Result |
|---|---|---|---|---|---|
| Latif et al. [25] | FAU-AIBO, IEMOCAP, EMO-DB, EMOVO, SAVEE | German, English, Italian | **Positive valence:** Surprise, motherese, joyful/happy, neutral, rest, excited, **Negative valence:** Angry, touchy, sadness, emphatic, reprimanding, boredom disgust, fear | eGeMAPS + DBN + Leave-One-Out | 80.00% (Accuracy) |
| Latif et al. [36] | EMOVO, URDU, SAVEE, EMO-DB | British English, Italian, German, Urdu | **Positive valence:** Anger, sadness, fear, boredom, disgust, **Negative valence:** Neutral, happiness, surprise | eGeMAPS + SVM | 70.98% (Accuracy) |
| Ocquaye et al. [37] | FAU-Aibo, IEMOCAP, EMO-DB, EMOVO, SAVEE | German, English, Italian | **Positive valence:** Surprise, motherese, joyful/happy, neutral, rest, excited, **Negative valence:** Angry, touchy, sadness, emphatic, reprimanding, boredom disgust, fear | triple attentive asymmetric CNN model | 73.11% (Accuracy) |
| Mustaqeem et al. [38] | RAVDESS, IEMOCAP | North American English, | Anger, happy, neutral, sad | Clean spectrograms + DSCNN | 56.50% (Accuracy) |
| Liu et al. [39] | eNTERFACE, EmoDB | English | Angry, disgust, fear, happy, sad, and surprise | DALSR + SVM | 52.27% (UAR) |
| Li et al. [40] | FUJITSU, EMO-DB, CASIA | Japanese, German, Chinese | Neutral, happy, angry, sad | IS16 + MSF + LMT (logistic model trees) | 82.63% (F-Measure) |
| Parry et al. [41] | RAVDESS, IEMOCAP, EMO-DB, EMOVO, SAVEE, EPST | English, German, Italian | **Positive valence:** Elation, excitement, happiness, joy, pleasant surprise, pride, surprise, **Negative valence:** Anger, anxiety, cold anger, contempt, despair, disgust, fear, frustration, hot anger, panic, sadness, shame, Neutral valence: Boredom, calm, interest, neutral | CNN | 55.11% (Average accuracy) |
| Deng et al. [42] | ABC, FAU AEC | German | **Positive valence:** Cheerful, neutral, rest, medium stress, **Negative valence:** Tired, aggressive, intoxicated, nervous, screaming, fear, high stress | INTERSPEECH 2009 Emotion Challenge baseline feature set + A-DAE (adaptive denoising autoencoders) | 64.18% (UAR) |
| Proposed Model | EMO-DB, SAVEE, RAVDESS, EMOVO, CREMA-D | German, British English, North American English | **Positive valence:** Anger, sadness, fear, disgust, boredom, **Negative valence:** Neutral, happiness, surprise | RF-RFE extracted features + RALOG | 96.60% (Accuracy) 96.00% (Recall) |

Ocquaye et al. [37] proposed a novel triple attentive asymmetric CNN model to recognize emotion using the cross-language approach. Their model achieved an accuracy score of 73.11% after training on IEMOCAP and tested on the FAU Aibo corpus. Mustaqeem et al. [38] achieved an overall accuracy of 56.5% using clean spectrograms with deep stride convolutional neural network (DSCNN) learning algorithm. They used a Ryerson audio-visual database of emotional speech and song (RAVDESS) and IEMOCAP speech emotion corpora to train and test their proposed model across four emotional states of anger, happy, neutral, and sad. The domain-adaptive least squares regression (DALSR) with SVM learning algorithm was presented and tested on eNTERFACE audio-visual emotion dataset and EMO-DB [39]. The study was done across six emotional states of angry, disgust, fear, happy, sad, and surprise to achieve an unweighted average accuracy (UAR) of 52.27%. Li et al. [40] combined an IS16 set of prosodic features with a speech modulation spectral feature (MSF) to develop a cross-language speech emotion recognition model [40]. They used the logistic model trees (LMT) learning algorithm to recognize four emotional states of neutral, happy, angry, and sad. The training and testing were done using the Fujitsu, EMO-DB and Chinese emotional database (CASIA) speech emotion corpora to achieve an F-Measure score of 82.63%. Speech emotion corpora of RAVDESS,

IEMOCAP, EMO-DB, Italian emotional speech database (EMOVO), SAVEE, and emotional prosody speech and transcript (EPST) were used to train and test a cross-language speech emotion recognition model. The emotional states in these corpora were divided into three sentiment groups of positive, negative, and neutral. Deng et al. [42] used an INTERSPEECH 2009 emotion challenge baseline feature set and adaptive denoising autoencoder (A-DAE) to develop a cross-language speech emotion recognition model. They trained and tested their proposed model on two German emotion corpora of airplane behaviors corpus (ABC) and FAU Aibo to achieve an unweighted average recall of 64.18%.

The review of the related literature has indicated in this paper that relatively few studies have been reported on cross-language speech emotion recognition. Many of these few studies have focused on the fusion of three languages and have reported performance results that need further improvement. The authors of the present study have investigated the problem of cross-language valence speech emotion recognition for different languages documented in the five corporal of EMO-DB, SAVEE, RAVDESS, EMOVO, and crowd-sourced emotional multimodal acted dataset (CREMA-D). The application of the RF-RFE algorithm for acoustic feature extraction following an existing study and validated with the RALOG ensemble learning algorithm gave the state-of-the-art results of 96.60% accuracy and 96.00% recall. The results have reflected significant improvement over the highest benchmarked result of 80.00% reported in the literature [25], as shown in Table 1.

## 3. Materials and Methods

The methodology of this research work follows a 5-phase cross-language valence emotion recognition process of the fusion of various speech emotion corpora, feature extraction, feature selection, ensemble creation, and recognition of valence. The first phase involves the fusion of five speech emotion corpora of EMO-DB, SAVEE, RAVDESS, EMOVO, and CREMA-D that constitute the materials for this study. These speech emotion corpora were deliberately chosen because they consist of speech files spoken in different languages and accents of North American English (RAVDESS), British English (SAVEE), German (EMO-DB), Italian (EMOVO), and a variety of other English accents (CREMA-D). The phase was subsequently followed by the feature extraction process to extract discriminating acoustic features that would help to improve the performance of the proposed RALOG ensemble algorithm. The extracted features include two main categories of prosodic and spectral features because of their proven effectiveness for the problem of single-language speech emotion recognition. The feature extraction phase was thereafter followed by a feature selection process to reduce the dimensionality that can ultimately impact both the training time and accuracy of the proposed algorithm. This phase was subsequently followed by ensemble creation based on statistical analysis of the correlation of receiver operating curve (ROC) and area under the curve (AUC) scores. The learning algorithms are logistic regression (LR), decision tree (DT), random decision forest (RDF), gradient boosting machine (GBM), multi-layer perceptron (MLP), support vector machine (SVM), K-nearest-neighbor (KNN), and AdaBoost with random forest (ARF).

The last phase of the study methodology concerns the actual recognition of valence emotion with a comparative evaluation of the proposed RALOG algorithm against four other existing ensemble algorithms. All experiments were implemented on a computer with an i7 2.3 GHz processor and 8 GB of RAM to test the effectiveness of the ensemble algorithms investigated in this study.

### 3.1. Speech Emotion Corpora

The speech emotion corpora of the present study are EMO-DB, SAVEE, RAVDESS, EMOVO, and CREMA-D. The EMO-DB was developed using ten professional German language speaking actors who were five males and five females. Their German spoken utterances were recorded to express seven emotional states of happiness, boredom, disgust, sadness, anger, neutral, and fear. The SAVEE corpus consists of 480 utterances spoken in the British English accent [19]. The corpus was developed using 4 male professional actors expressing seven emotional states of angry, disgust, fear, happy, neutral, sad, and surprise. The RAVDESS is a gender-balanced emotion corpus that consists of 1440

vocal utterances that were recorded from professional actors involving twelve males and twelve females [43]. The corpus was developed with the North American English accent, which is different from the one used in SAVEE. Moreover, RAVDESS is a multiclass emotion corpus that consists of eight emotional states of angry, calm, disgust, fear, happy, neutral, sad, and surprise. The EMOVO corpus is an Italian emotional speech database that consists of utterances spoken in the Italian language [44]. The corpus was developed using six professional actors involving 3 males and 3 females who simulated seven emotional states of disgust, fear, anger, joy, surprise, sadness, and neutral. The CREMA-D corpus constitutes 6786 audio-visual clips recorded from 91 professional actors involving 48 males and 43 females from diverse races and ethnic backgrounds, including African, American, Asian, Caucasian, Hispanic, and Unspecified [45].

In this study, only the audio files of five speech emotion databases were fused to create a cross-language experimentation corpus as inspired by the previous studies [25,40]. The databases were specifically chosen because they consist of vocal utterances spoken in different languages and varying accents. The cross-language corpus was designed using eight speech emotional states of angry, happy, neutral, calm, sad, surprised, fear, and disgust. These emotions were mapped to positive and negative valences, as illustrated in Table 2 [22,25,41]. The mapping of speech emotions to emotional valences has provided a seamless mechanism to unify different corpora with varying emotional states for cross-language emotion recognition experimentation. The distribution of utterances across the five corporal indicates a total of 10,094 adult utterances for the experimental dataset of this study. The distribution by gender categorization is imbalanced because about 54% of the total utterances constituted male utterances, while about 46% are female utterances. The SAVEE corpus contains male utterances, and the CREMA-D contains more male utterances than female utterances, which have made the experimental dataset imbalanced. The training and testing datasets of utterances respectively constituted 80% and 20% of the total utterances.

**Table 2.** Mappings of adult speech emotions into positive and negative valence emotions.

| Corpus | Language | Total Utterances | | Training Utterances | | Testing Utterances | | Negative Valence | Positive Valence |
|---|---|---|---|---|---|---|---|---|---|
| | | Male | Female | Male | Female | Male | Female | | |
| EMO-DB | German | 400 | 400 | 320 | 320 | 80 | 80 | Anger, Sadness, Fear, Disgust, Boredom | Neutral, Happiness |
| SAVEE | British English | 480 | 0 | 384 | 0 | 96 | 0 | Anger, Sadness, Fear, Disgust | Neutral, Happiness, Surprise |
| RAVDESS | North American English | 720 | 720 | 576 | 576 | 144 | 144 | Anger, Sadness, Fear, Disgust | Neutral, Happiness, Surprise, Calm |
| EMOVO | Italian | 294 | 294 | 235 | 235 | 59 | 59 | Anger, Sadness, Fear, Disgust | Neutral, Happiness, Surprise |
| CREMA-D | African, American, Asian, Caucasian, Hispanic, and Unspecified | 3579 | 3207 | 2863 | 2566 | 716 | 641 | Anger, Sadness, Fear, Disgust | Neutral, Happiness |

The resulting cross-language corpus has produced 6641 sample files of negative valences, and 453 sample files present positive valences. It is important to observe that the dominance of negative valence emotions over positive valence emotions did not bias the prediction results toward the negative valence emotions. Figure 1 shows the distribution of both the negative valence instances (unsatisfied class) and positive valence instances (satisfied class) that are presented in the cross-language corpus of this study.

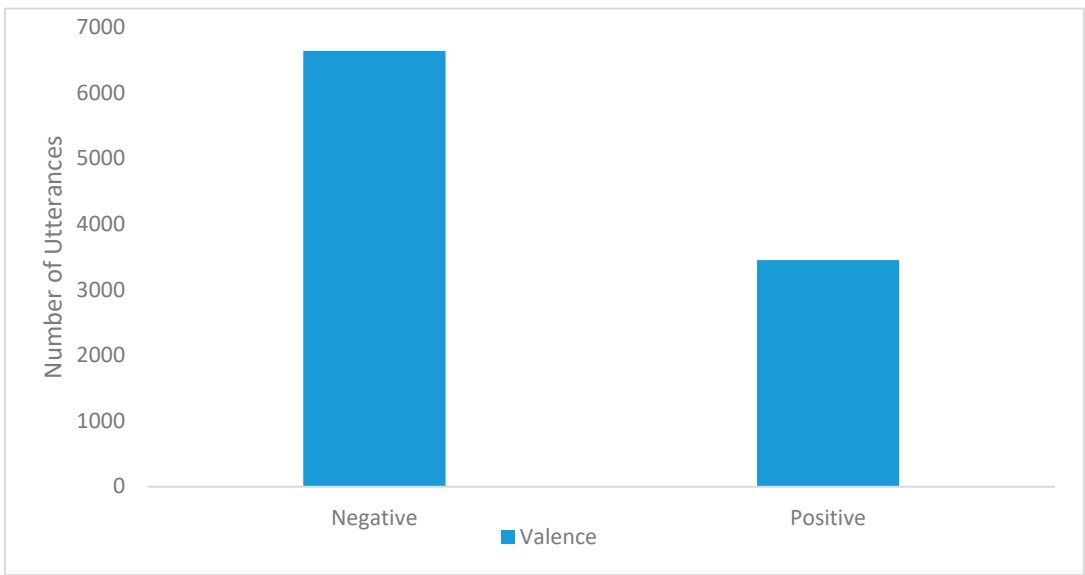

**Figure 1.** Combined speech emotion dataset.

*3.2. Feature Extraction*

The process of feature extraction is an important component of speech processing architecture because feature vectors are traditionally fed as input to a speech recognition system. The extraction of a set of discriminating features is considered essential for improving the performance of a valence emotion recognition system. The extraction of hybrid acoustic features (HAF) from the raw audio files in the cross-language corpus has been performed. HAF features were considered for this study because of their excellent performance in an experimental case of a single-language corpora [46]. A lot of features have been proposed in the literature, but most of the proposals were silent on the ranking of those features that are highly discriminating. The general lack of feature ranking in a multilingual speech corpus presents a huge void to speech processing applications. A total of 404 spectral and prosodic speech features as shown in Table 3 were extracted using jAudio [47], which is an open source feature extraction tool developed using the Java programming language.

**Table 3.** List of 404 hybrid acoustic features (HAF) features [46].

| Group | Type | Number |
|---|---|---|
| **Prosodic** | | |
| Energy | Logarithm of Energy | 10 |
| Pitch | Fundamental Frequency | 70 |
| Times | Zero Crossing Rate | 24 |
| **Spectral** | | |
| Cepstral | MFCC | 133 |
| Shape | Spectral Roll-off Point | 12 |
| Amplitude | Spectral Flux | 12 |
| Moment | Spectral Centroid | 22 |
| Audio | Spectral Compactness | 10 |
| Frequency | Fast Fourier Transform | 9 |
| Signature | Spectral Variability | 21 |
| Envelope | LPCC | 81 |

*3.3. Feature Selection*

Feature selection is one of the most crucial steps in the development of efficient pattern recognition models [48]. We observed that some models reported in the literature have yielded low accuracy scores and presented high training times. The consequence is that this may present unwarranted hiccups when

a proposed model is deployed for the purpose of real-time commercial applications [49]. The inherent problem would persist severely in multilingual speech recognition systems because of the different styles of expressing satisfaction or dissatisfaction across various languages, cultures, and accents. Consequently, we have decided to perform feature selection using the RF-RFE algorithm, which has been reported to be successful in gender voice recognition [31]. Figure 2 depicts the iterative cycle involved in the application of the RF-RFE algorithm.

**Inputs:**
    Training set Tr
    Set of $\alpha$ features Fe= $\{f_1.........f_\alpha\}$
    Ranking method M (Tr, Fe)
**Outputs:**
    Final ranking $R$
    Code:
    Repeat for $i$ in $\{1: \alpha\}$
    Rank set Fe using M (Tr, Fe)
    $f^* \leftarrow$ last ranked feature in Fe
    $R (\alpha - i + 1) \leftarrow f^*$
    Fe$\leftarrow$ Fe $- f^*$

**Figure 2.** Summarized random forest recursive feature elimination (RF-RFE) algorithm.

After running the RF-RFE algorithm on the cross-language corpus of this study, 250 features were chosen as the most discriminating. The top 15 discriminative features were ranked and presented in Figure 3. The application of the RF-RFE algorithm has reduced the complexity of using a large set of features because not all the acoustic features proposed in [46] were used in this study; instead, we used a highly discriminating subset.

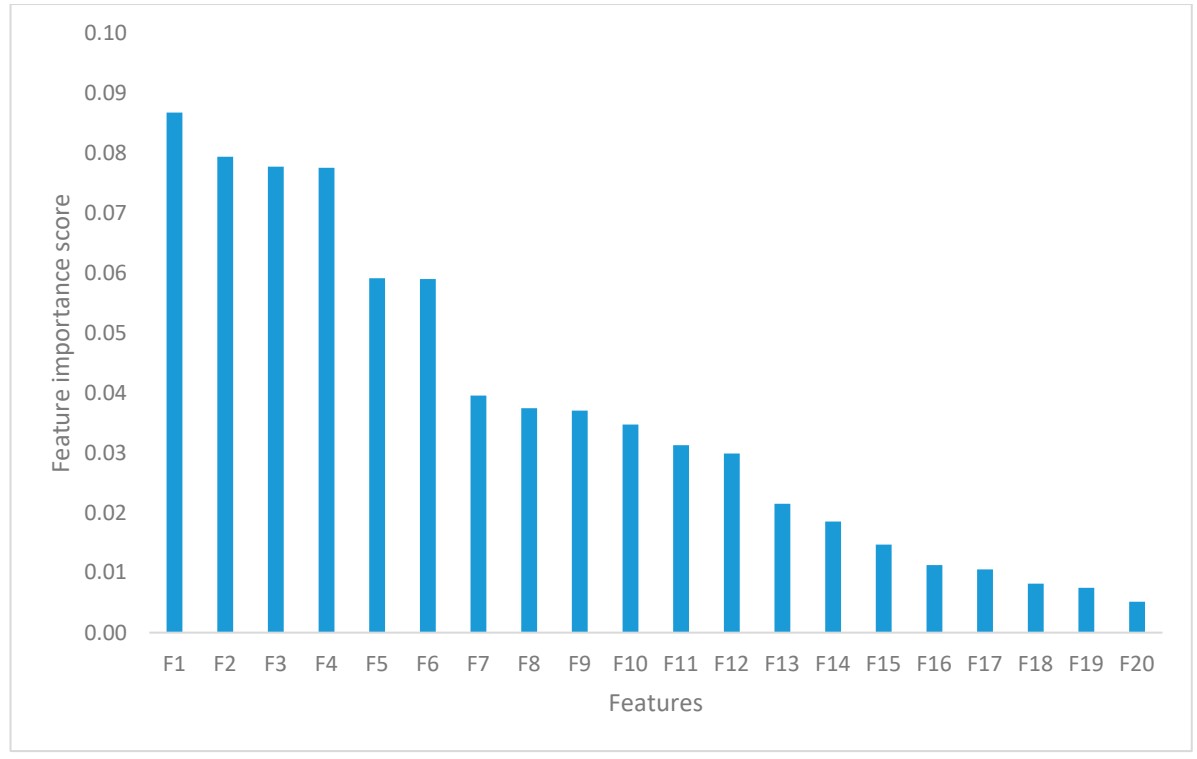

**Figure 3.** Top 20 RF-RFE ranked acoustic features.

Table 4 presents the descriptions of the top 20 highly discriminative acoustic features from the HAF set, as shown in Figure 3. Accordingly, the derivative of standard deviation of area method of moments of overall standard deviation (F1) is the feature that was ranked highest among the entire acoustic features.

**Table 4.** Description of the top 20 acoustic features.

| Key | Feature Description |
|-----|---------------------|
| F1  | Derivative of standard deviation of area method of moments overall standard deviation (6th variant) |
| F2  | Derivative of standard deviation of relative difference function overall standard deviation standard (1st variant) |
| F3  | Area method of moments overall standard deviation (7th variant) |
| F4  | Peak detection overall average (10th variant) |
| F5  | Standard deviation of area method of moments overall standard deviation (8th variant) |
| F6  | Derivative of area method of moments overall standard deviation (8th variant) |
| F7  | Area method of moments of Mel frequency cepstral coefficients overall standard deviation (1st variant) |
| F8  | Derivative of area method of moments overall standard deviation (9th variant) |
| F9  | Derivative of standard deviation of area method of moments overall standard deviation (4th variant) |
| F10 | Standard deviation of method of moments overall average (2nd variant) |
| F11 | Peak detection overall average (10th variant) |
| F12 | Peak detection overall average (9th variant) |
| F13 | Peak detection overall average (7th variant) |
| F14 | Derivative of standard deviation of area method of moments overall standard deviation (6th variant) |
| F15 | Method of moments overall average (3rd variant) |

### 3.4. Ensemble Creation

The literature on machine learning has generally revealed the superiority of ensemble algorithms for speech recognition. Ensemble learning algorithms are well-known for yielding high recognition accuracy provided appropriate features are used to train the algorithms [50]. However, we have observed that most of the ensemble algorithms reported in the literature to date are either computationally expensive or they produce almost the same recognition accuracies as most of the base learners are sometimes called inducers [46,51]. Hence, we have decided to develop a new ensemble algorithm using the statistical analysis of correlation of ROC AUC score to nominate the learning algorithms that would become part of the proposed RALOG ensemble algorithm. The learning algorithms used at this phase are decision tree, random decision forest, gradient boosting machine, logistic regression, multi-layer perceptron, support vector machine, k-nearest-neighbor, and AdaBoost with random forest. The correlation matrix computed using the ROC AUC scores of these learning algorithms is illustrated in Figure 4.

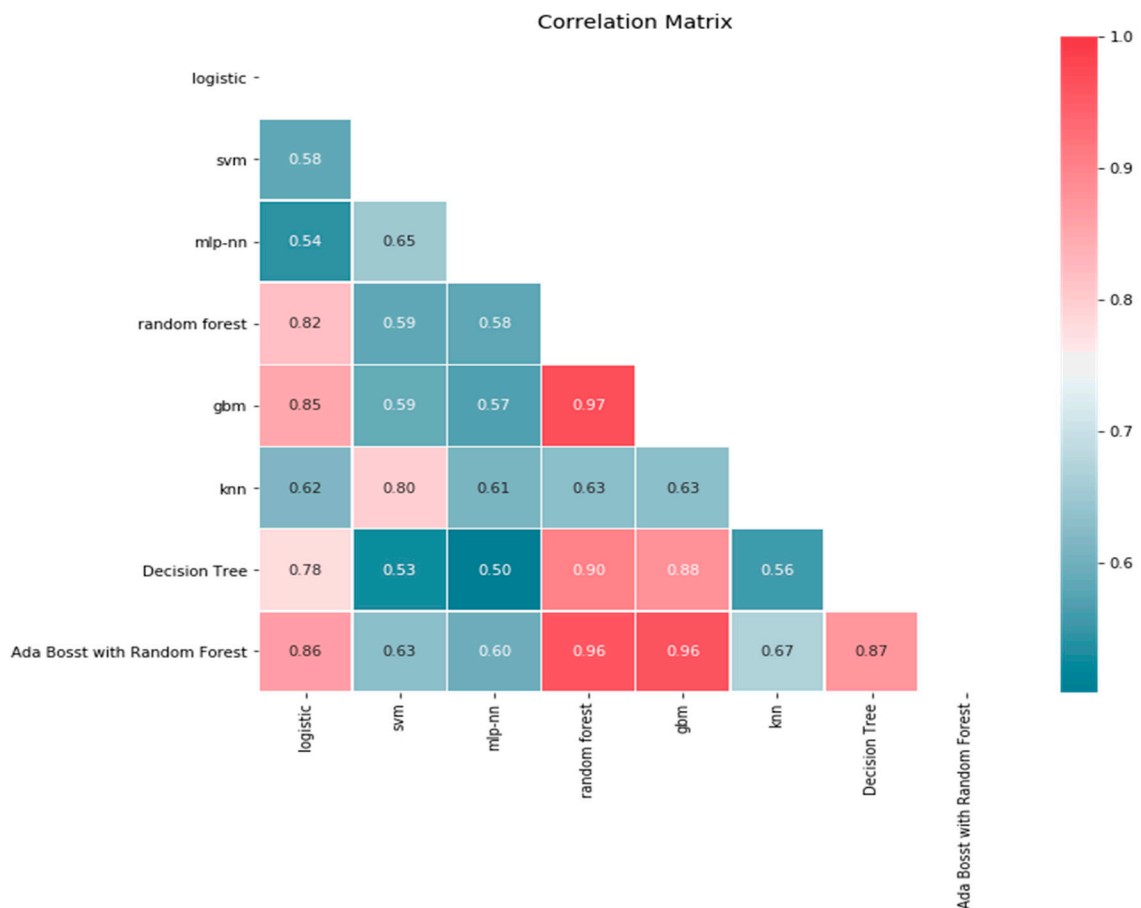

**Figure 4.** Correlation matrix of learning algorithms.

The approach of stacking ensemble was applied to develop the proposed RALOG ensemble algorithm using the mixtend package [52]. The package is a publicly available library of useful python tools for developing ensemble learning algorithms. The AdaBoost with random forest was used as a meta-learner for RALOG because it has yielded a high score in the correlation matrix. Logistic regression, decision tree, random decision forest, gradient boosting machine, and AdaBoost with random forest were the experimental learning algorithms for RALOG. They have high total correlation scores and are chosen based on the correlation matrix shown in Figure 4. The architecture of the RALOG stacked ensemble algorithm is illustrated in Figure 5.

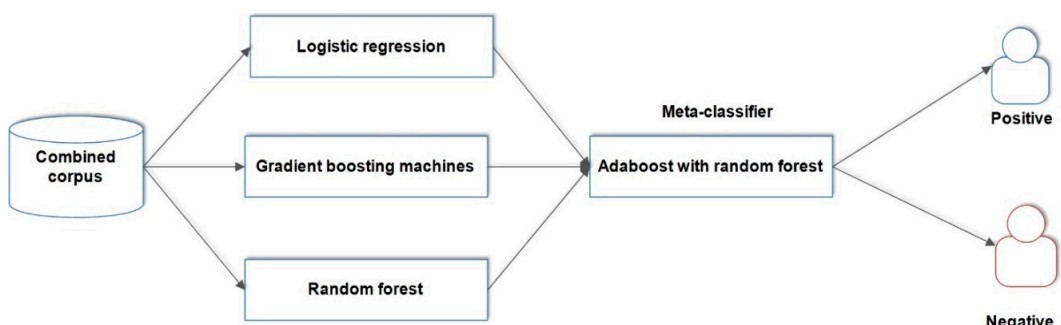

**Figure 5.** Architecture of RALOG ensemble learning algorithm: a combination of random decision forest, AdaBoost, logistic regression, and gradient boosting machine.

Algorithm 1 shows the pseudocode of the RALOG stacked ensemble algorithm where $\chi = \{x_i \in R^M\}$, $\Upsilon = \{y_i \in N\}$, and a training set $D = \{(x_i, y_i)\}$. The parameters of the RALOG

algorithm are that $\chi$ represents the data without the labels, $x_i$ represents an instance in a dataset, $\Upsilon$ represents a collection of labels from the dataset, and $R^M$ represents a pool of instances in a dataset. $S$ is the number of algorithms to be used to develop the stacked ensemble, $x^{new}_i$ represents a new instance in a newly developed testing dataset. The dataset is a surrogate of D, the original dataset, $a_s$ represents the learner-based algorithms, and $a^{new}$ represents the new stacked ensemble algorithms developed using $(a_1(x), a_2(x), \ldots, a_S(x))$ that define a list of algorithms that are stacked together.

---

**Algorithm 1: Stacking Ensemble Algorithm**

---

**Input:** D = {$(x_i, y_i)| x_i \in \chi, y_i \in \Upsilon$ }
**Output: An ensemble algorithm A**
**Step 1:** Learn first-level learning algorithms
For $s \leftarrow 1$ to $S$ do
Learn a base learning algorithm $a_s$ based on D
**Step 2:** Construct a new dataset from D
For $i \leftarrow 1$ to m do
Construct a new dataset that contains {$x^{new}_i, y_i$}, where
$x^{new}_i = \{a_j(x_i)$ for $j = 1$ to $S\}$
end
**Step 3:** Learn a second-level learning algorithm
Learn a new learning algorithm $a^{new}$ based on the newly constructed dataset
**Return** A$(x_i) = a^{new} (a_1(x), a_2(x), \ldots, a_S(x))$

---

### 3.5. Recognition of Valence

The prime purpose of valence speech emotion recognition was to obtain the valence state of an input vector to a learning algorithm. In this study, several experiments were performed to concomitantly evaluate the effectiveness of the scaled-down version of HAF features as input to the RALOG algorithm. The RALOG algorithm was tested against other state-of-the-art algorithms, which are extra gradient boosting machine (XGB), random decision forest (RDF), extra trees (ETC), and gradient boosting machine (GBM). Ensemble algorithms are normally used to improve the recognition capability of input features by joining the predictions of several learning algorithms [53]. They boost the predictive accuracy by combining a set of weak learners [54]. In this study, we have applied stacking, boosting, and bagging ensemble learning algorithms, because they have achieved excellent results in other works [31].

RALOG is a stacking ensemble used in this study, while XGB and GBM are the boosting ensemble algorithms [31], and the bagging ensemble algorithms are RDF and ETC [31] because they were reported as successful in previous studies. The training time, accuracy, precision, recall, and F1-score are the standard metrics used to evaluate the performance of the proposed RALOG algorithm. Furthermore, confidence intervals were used because the experimental data were not well balanced. Figure 6 shows the flowchart for different configurations of varying experiments that were conducted in this study.

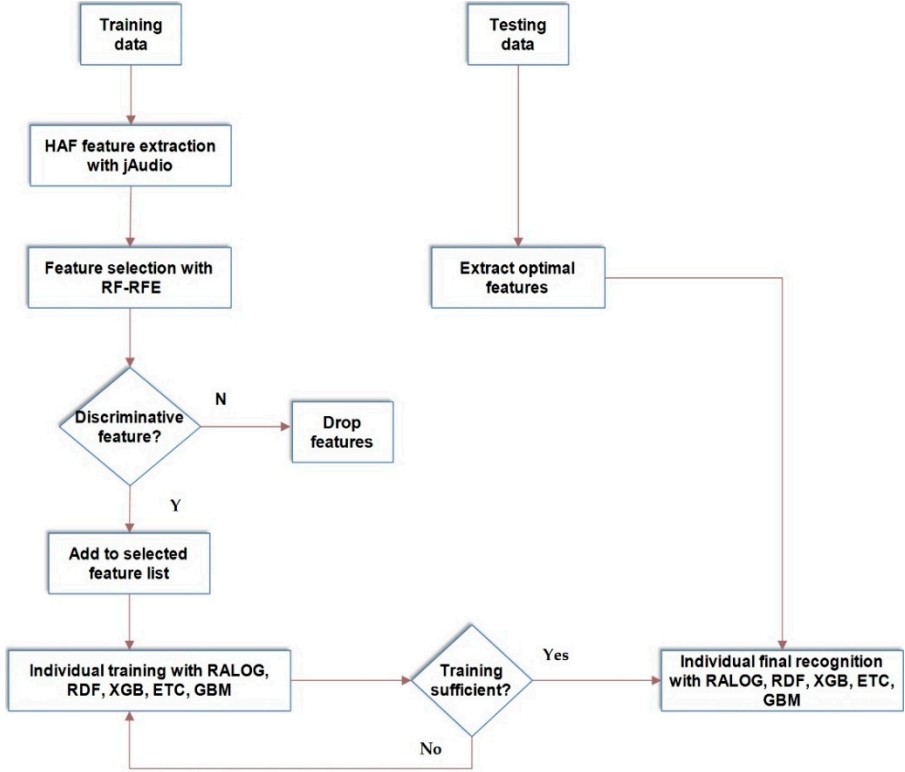

**Figure 6.** Different configurations of valence recognition experiments derived from [46].

## 4. Results and Discussion

In this study, negative and positive valence emotion files were extracted from the speech corpora of EMO-DB, SAVEE, RAVDESS, EMOVO, and CREMA-D to create the cross-language corpus according to the mapping in [25]. Thereafter, 250 of the most discriminating HAF features were selected from the speech files for the assignment of valence speech emotion recognition. The effectiveness of the proposed RALOG algorithm was compared with other popular ensemble algorithms trained to learn the extracted HAF features. Cross-language speech data used for experimentation were subjected to 10-fold cross-validation and divided into training and testing groups as the standard norm permits. Inthe experimentation, 20% of the data (sets of adult utterances) were used for testing, while 80% of the data were used for training as previously mentioned.

Table 5 shows the central processing unit (CPU) time of training the individual ensemble algorithm investigated in this study. The GBM algorithm was seen to be the most time-consuming, which was greatly reduced after performing feature selection. It took 60.495 ms to train the GBM algorithm before running the RF-RFE algorithm, but it only took 16.228 ms after applying the feature selection algorithm. This pattern was consistent for all the other learning algorithms, including the proposed RALOG. In addition, the results show that the RALOG algorithm had a healthy training time of 9.05 ms. However, the XGB algorithm proved to be the fastest amongst the learning algorithms because it only took 0.00095 ms to process data. These results imply that RALOG is highly promising for recognizing valence speech emotion, especially in the situation where reactionary delay is of crucial importance [55]. Reactionary delay is an important factor in the development of real-time systems.

**Table 5.** Training time (ms) of ensemble algorithms.

|            | RALOG  | RDF   | GBM    | ETC   | XGB   |
|------------|--------|-------|--------|-------|-------|
| Before RFE | 43.314 | 5.290 | 60.495 | 0.168 | 0.004 |
| After RFE  | 9.050  | 0.459 | 16.228 | 0.244 | 0.001 |



The experimental results obtained in this study in terms of precision, recall, F1-score, and accuracy are illustrated in Table 6. The results show that feature selection with the aid of the RF-RFE algorithm contributed heavily to a significant increase in the recognition accuracy. The recognition accuracy of RALOG increased from 82% to 92% while that of XGB algorithm increased from 78% to 87%. The increase in accuracy was apparent in all results of the learning algorithms. However, the highest increment was observed when the RF-RFE algorithm was used with the ETC learning algorithm, which has resulted in a 10% increment. Even though the ETC learning algorithm recorded the highest increment, it has yielded a lower accuracy score when compared to other learning algorithms. It can be observed from results in Table 6 that the RDF algorithm has performed well in recognizing valence speech emotions because it achieved a recognition accuracy score of 85%. In the same vein, the results show that the ETC algorithm is a good option for recognizing valence speech emotion because it achieved a recognition accuracy of 83% after applying the RF-RFE algorithm. RALOG achieved the highest accuracy before and after the application of the RF-RFE feature selection algorithm. It even performed better when compared to the other algorithms presented in the literature. For instance, Latif et al. [25] achieved an overall accuracy of 80% using eGeMAPS, DBN, and leave-one-out testing technique. Hence, it can be inferred that RALOG is highly promising for recognizing valence speech emotions, especially when it is used in conjunction with the RF-RFE algorithm. Moreover, the confidence factor shows that the true recognition accuracies of all learning algorithms lie in the range of 0.005% and 0.009%.

**Table 6.** Percentage average precision, recall, F1-score, and accuracy with confidence intervals of learning algorithms at each experimental stage of feature scaling with the RF-RFE algorithm.

| Algorithm | Stage | Precision | Recall | F1-Score | Accuracy |
|---|---|---|---|---|---|
| **RALOG** | **Before RFE** | 82 (±0.007) | 83 (±0.007) | 83 (±0.007) | 82 (±0.007) |
| | **After RFE** | 90 (±0.006) | 90 (±0.006) | 91 (±0.006) | 92 (±0.005) |
| **RDF** | **Before RFE** | 72 (±0.009) | 82 (±0.007) | 77 (±0.008) | 80 (±0.008) |
| | **After RFE** | 80 (±0.008) | 88 (±0.006) | 82 (±0.007) | 85 (±0.007) |
| **GBM** | **Before RFE** | 69 (±0.009) | 73 (±0.009) | 71 (±0.009) | 73 (±0.009) |
| | **After RFE** | 79 (±0.008) | 82 (±0.007) | 80 (±0.008) | 85 (±0.007) |
| **XGB** | **Before RFE** | 71 (±0.009) | 81 (±0.008) | 75 (±0.008) | 78 (±0.008) |
| | **After RFE** | 78 (±0.008) | 85 (±0.007) | 81 (±0.006) | 87 (±0.007) |

The same pattern of performance was observed in the precision, recall, and F1-score of the ensemble algorithms. The superiority of the proposed RALOG algorithm was also observed when precision was analyzed in Table 6. The impact of using the RF-RFE algorithm to scale features was again noticed when the precision was evaluated. The average precision for all the learning algorithms increased considerably. GBM experienced the highest increment from 69% to 79%, while the average precision of RALOG increased from 82% to 90%. The RALOG algorithm achieved the highest average precision of 90% followed by the RDF algorithm (80%). The RDF algorithm was closely followed by the GBM algorithm that achieved an average precision score of 79%. In this regard, the GBM was followed by the ETC and XGB algorithms that achieved similar average precision scores of 78% each. It can be observed from the results in Table 6 that RALOG is a good learning algorithm for recognizing valence emotion in speech files, and the results are comparable to those of the work presented in [39].

The average recall as illustrated in Table 6 was improved marginally when feature selection was applied. We observed that RALOG outperformed the other learning algorithms in this regard because it obtained an average recall score of 90%. RALOG was followed by RDF (88%), XGB (85%), GBM (82%), and ETC (82%). Although GBM achieved the lowest average recall score, it had the highest percentage increment of 11% after feature scaling. These results present a further confirmation that strongly underlines the fact that the application of RALOG with the RF-RFE algorithm for valence emotion recognition is indeed promising. The average recall of RALOG is relatively high when compared to the

results in the literature [39]. In addition, RALOG achieved the highest average F1-score of 91% followed by RDF (82%) and XGB (81%). ETC and GBM both achieved the lowest average F1-score of 80%. All the learning algorithms increased their F1-scores for each respective class when the RF-RFE algorithm was applied. Furthermore, ETC and GBM achieved the highest average F1-score increment of 9%. Moreover, the results confirm that the performance of RALOG is comparable to the proposal presented in the literature [56].

Table 7 shows the results of percentage precision, recall, F1-score, and accuracy for valence instances. The results show that ensemble algorithms achieved lower scores in recognizing instances of valence emotion. This could be ascribed to the unbalanced structure of the cross-language corpus. There was a significant increase in performance across all performance metrics when the RF-RFE algorithm was applied. Specifically, RDF achieved the highest precision score of 97% in recognizing negative valence emotion instances followed by XGB (95%). RALOG and ETC both achieved a precision score of 93% in recognizing instances of negative valence. However, RALOG achieved the highest accuracy score of 97% in recognizing instances of negative valence emotion. Furthermore, RALOG had the highest precision score of 87% in recognizing instances of positive valence.

**Table 7.** Percentage precision, recall and F1-score and accuracy at each experimental stage of feature scaling with the RF-RFE algorithm.

| Algorithm | Stage | Precision | | Recall | | F1-score | | Accuracy | |
|---|---|---|---|---|---|---|---|---|---|
| | | Negative | Positive | Negative | Positive | Negative | Positive | Negative | Positive |
| **RALOG** | **Before RFE** | 88 | 76 | 86 | 79 | 88 | 77 | 86 | 78 |
| | **After RFE** | 93 | 87 | 94 | 86 | 94 | 87 | 97 | 86 |
| **RDF** | **Before RFE** | 83 | 59 | 84 | 78 | 82 | 67 | 82 | 74 |
| | **After RFE** | 97 | 86 | 97 | 62 | 88 | 87 | 92 | 72 |
| **GBM** | **Before RFE** | 81 | 56 | 78 | 64 | 80 | 62 | 79 | 71 |
| | **After RFE** | 92 | 66 | 89 | 74 | 90 | 70 | 91 | 79 |
| **XGB** | **Before RFE** | 84 | 60 | 85 | 79 | 83 | 71 | 84 | 76 |
| | **After RFE** | 95 | 61 | 88 | 82 | 91 | 70 | 94 | 80 |

The results in Table 8 confirm the effectiveness of applying the RF-RFE algorithm for cross-language valence speech emotion recognition. These results show a significant improvement in valence emotion recognition because they are comparable to the work presented in Table 1.

**Table 8.** Confusion matrix before and after feature scaling with the RF-RFE algorithm.

| Algorithm | Valence | Before RFE | | After RFE | |
|---|---|---|---|---|---|
| | | Negative | Positive | Negative | Positive |
| **RALOG** | **Negative** | 88 | 12 | 93 | 7 |
| | **Positive** | 24 | 76 | 13 | 87 |
| **RDF** | **Negative** | 83 | 17 | 97 | 3 |
| | **Positive** | 41 | 59 | 38 | 62 |
| **GBM** | **Negative** | 81 | 19 | 92 | 8 |
| | **Positive** | 41 | 59 | 34 | 66 |
| **ETC** | **Negative** | 80 | 20 | 93 | 7 |
| | **Positive** | 43 | 57 | 37 | 63 |
| **XGB** | **Negative** | 84 | 16 | 95 | 5 |
| | **Positive** | 40 | 60 | 39 | 61 |

The ROC curves of the evaluated learning algorithms before and after applying the RF-RFE algorithm are shown in Figures 7 and 8, respectively. Figure 7 shows the performance of the ensemble algorithms before the feature selection was done. It can be noted from the ROC curves that RALOG outperformed all the other algorithms before and after the RF-RFE algorithm was applied. RALOG had the highest AUC score of 96.97% when the RF-RFE algorithm was applied. This means that there is a 96.97% chance that correct unpleasant valence emotion will be recognized using RALOG with the RF-RFE algorithm. It was closely followed by GBM, which achieved an AUC score of 94.62%. These results show that the selected features by the RF-RFE algorithm can faithfully recognize valence speech emotion from different languages, accents, and recording environments because the corpus used for experimentation in this study was a fusion of five different emotion corpora.

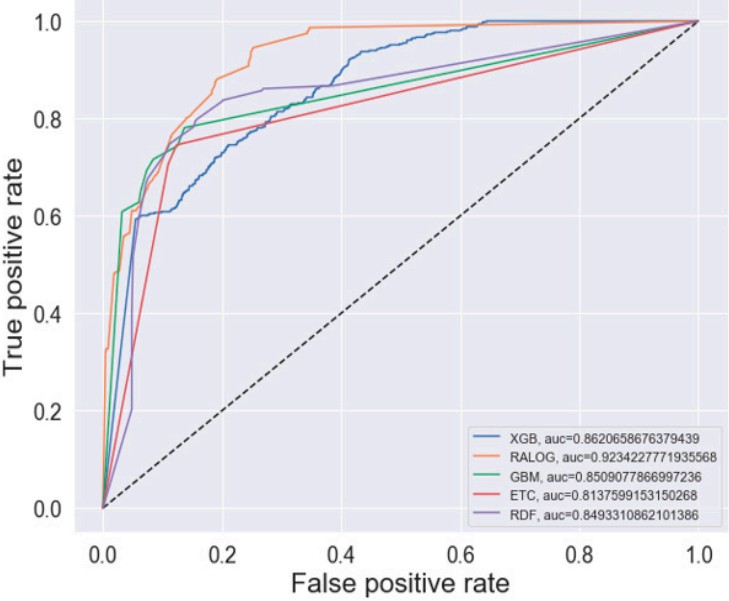

**Figure 7.** Receiver operating curve (ROC) curve before recursive feature elimination (RFE).

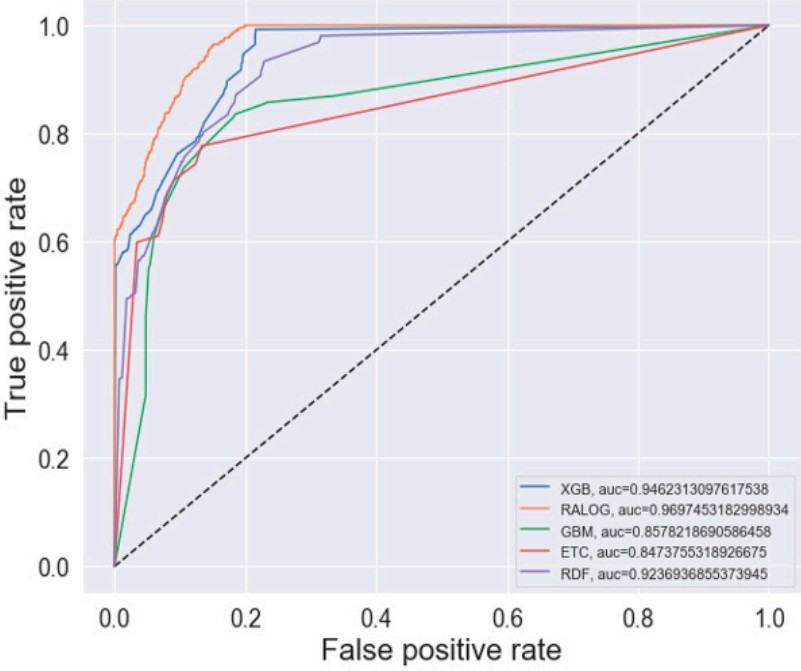

**Figure 8.** ROC curve after RFE.

The final experiment was performed to determine the performance results of the RALOG algorithm on an individual corpus. Table 9 shows these results, wherein it can be seen that the recognition results are relatively good across corpora. Specifically, RALOG gave the best recognition performance results for SAVEE, because an overall accuracy of 98% was recorded. The best performance of RALOG on SAVEE is because the corpus only consists of male utterances from Britain. A reduction in the recognition performance results is observed as we investigate the gender-balanced corpora. However, the lowest recognition performance results were noticed across the four metrics when RALOG was applied on CREMA-D. This is because CREMA-D consists of utterances from diverse accents of speakers with multiple cultures, which confirms the difficulties experienced in recognizing valence emotion in a multilingual environment, as reported in the literature [21,22]. The recognition results of RALOG on the agglutination of the five corpora are not better than its results on an individual SAVEE corpus and RAVDESS corpus, but they are better than its results on EMOVO and CREMA-D corpora. Consequently, male valence emotion recognition performance reaches the best results on the SAVEE corpus, but the performance on the combined corpora is generally more impressive irrespective of gender than when an individual corpus is considered.

**Table 9.** Percentage precision, recall and F1-score and accuracy of RALOG on individual corpus at each experimental stage of feature scaling with the RF-RFE algorithm.

| Corpus | Stage | Precision | Recall | F1-Score | Accuracy |
|--------|-------|-----------|--------|----------|----------|
| **EMO-DB** | **Before RFE** | 95 | 96 | 95 | 95 |
|  | **After RFE** | 97 | 96 | 96 | 96 |
| **SAVEE** | **Before RFE** | 97 | 96 | 96 | 96 |
|  | **After RFE** | 98 | 98 | 98 | 98 |
| **RAVDEES** | **Before RFE** | 94 | 94 | 94 | 95 |
|  | **After RFE** | 97 | 97 | 97 | 97 |
| **EMOVO** | **Before RFE** | 92 | 92 | 93 | 92 |
|  | **After RFE** | 94 | 94 | 94 | 95 |
| **CREMA-D** | **Before RFE** | 88 | 89 | 88 | 89 |
|  | **After RFE** | 85 | 95 | 89 | 92 |

Table 10 shows the results of the training time of RALOG on individual corpus. The algorithm is seen generally to have recorded the lowest training time of 0.47 ms on SAVEE and the highest training time of 6.05 (ms) on CREMA-D. This result is obviously expected, because the least number of sample utterances that come from SAVEE for training RALOG is 384, which is less than those from CREMA-D (1358) and all other corpora.

**Table 10.** Training time (ms) of RALOG on individual corpus.

|  | **EMO-DB** | **SAVEE** | **RAVDEES** | **EMOVO** | **CREMA-D** |
|--|------------|-----------|-------------|-----------|-------------|
| Before RFE | 3.440 | 2.070 | 6.190 | 2.540 | 29.200 |
| After RFE | 0.780 | 0.470 | 1.380 | 0.580 | 6.050 |

## 5. Conclusions

The automatic recognition of valence emotion in speech is still an open research agenda because of various factors that include differences in the expression of emotions according to different cultures, accents, and languages. Limited research has been done in cross-language emotion recognition with an insufficient number of cross-corpora for research, as seen in the literature review section of this paper. The corpus of this study was developed by scaling down low-discriminating features from the HAF features that were constructed in the previous study through the agglutination of spectral and prosodic features. The feature scaling was done through the selection of highly discriminative features with the

aid of the RF-RFE algorithm. The scaling down of the redundant features has achieved excellent results in this study. The features were extracted from a cross-language speech emotion corpus that was developed in this study by fusing five different speech emotion corpora. The development of a RALOG stacked ensemble algorithm for recognizing cross-language valence emotion is an important highlight of this study. The algorithm was evaluated against other top performing learning algorithms, and it outperformed them before and after feature scaling. RALOG performed well because it combines strong learners to boost its performance. Furthermore, we have observed generally that ensemble algorithms perform well for the assignment of valence speech emotion recognition.

The results of this study are generally promising when compared to the results presented in the literature reviewed. The gradient boosting machine achieved good results, but the greatest bottleneck is that its training time is computationally expensive. However, XGB had the fastest training time, making it a good alternative for developing real-time valence emotion recognition systems. Even though the RALOG algorithm was not the fastest, it had a healthy training time and becomes a good candidate when considering the tradeoff between method efficiency and method accuracy. RALOG was seen to be the more accurate algorithm when compared to random decision forest algorithm in recognizing valence emotion through speech using a set of well-scaled HAF features. This superiority was consistent across all the ensemble algorithms. The results obtained are generally exhilarating; nonetheless, the core limitation of this study is that the cross-language corpora used did not contain conversations, but rather consisted of sentences per speaker. We would like to evaluate the performance of RALOG algorithm in actual conversations to fully scrutinize its robustness in future work. Although the data used in this study were free of noise, they were also devoid of speakers with physiological deficiencies such as stammering. As part of our future work, we would like to vigorously pursue this exciting research agenda.

**Author Contributions:** Conceptualization by K.Z. and O.O.O., Data curation by K.Z. Methodology by K.Z. Writing original draft by K.Z. and O.O.O. Writing review and editing by O.O.O., Supervision by O.O.O. All authors have read and agreed to the published version of the manuscript.

**Funding:** This research received no external funding.

**Acknowledgments:** The authors would like to sincerely thank the anonymous reviewers for the constructive sentiments and useful suggestions that have led to significant improvements in quality and presentation of this paper.

**Conflicts of Interest:** The authors declare no conflict of interest.

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
