# Peer review of "Recognition of Cross-Language Acoustic Emotional Valence Using Stacked Ensemble Learning"

_algorithms, doi:10.3390/a13100246_

Round 1
Reviewer 1 Report
Reviewer Comments / Remarks
General Comments
===============
The submitted paper deals with the problem of emotion recognition from speech using a fusion of multi- language corpora. In particular the authors propose the use of ensemble learning based on an algorithm that uses Random decision forest, Adaboost, LOgistic regression, and Gradient boosting machine (RALOG) presented in authors' previous works. Speech is analyzed to extract a list of spectral and procodic descriptors, and the RF-RFE algorithm was used to select the most important ones.
Experiments have shown that the proposed system can be used to succesfully classify negative / positive valence relatively quickly and accurately compared to other state of the art methods using a database consisting of five speech databases and speakers with four different languages.
Overall the quality of the paper is good, well-structured, with good use of English. The authors have key-pointed their contribution as well as the novelty of their proposed research. Experiments are extensive and the results have been compared to a number of well known valence recognition methods from the literature.
Specific Comments to the authors
=========================
1. Line 350. The authors should include the percentage splitting the have used for the training / validation / testing procedure in each one of the 10-fold cross validation procedure.
2. Lines 352-359. The authors should not relate the size of the stored features (in MB) after feature extraction with the "curse of dimensionality". The curse of dimensionality has to do with the number of the different descriptors that are used in the pattern recognition problem and not the demands for their storage on the disk.
3. Line 372. What exactly does the procesing time in msec that the authors have calculated and included in table 5 refer to? They should give more detail on what does (line 364) "time to process the data" mean.
4. Line 437. Have the authors used only 100 speech utterances to test their algorithm from 10094 utterances (line 229)? If so, they should test the performance of their algorithm using a much larger subset of the sample files with positive/negative valence values (~10% -> 1000 utteraces). What is the distribution of the different languages in the test samples that were used? How many of them belong to the four different languages (German, Italian, British English, American etc)? The bigger test set that should be used, must include utterances from all different corpora (uniformly distributed).
5. How do the paper's results and the efficacy of the proposed system compare to each different corpora on its own? Are the recognition results similar for each corpora and the four different speakers? Are any differences in the recognition accuracy in specific languages observed? The authors should definately investigate this and include their findings in the paper.
Author Response
Reviewer 1
Open Review
(x) I would not like to sign my review report
( ) I would like to sign my review report
English language and style
( ) Extensive editing of English language and style required
( ) Moderate English changes required
(x) English language and style are fine/minor spell check required
( ) I don't feel qualified to judge about the English language and style
|
Yes |
Can be improved |
Must be improved |
Not applicable |
|
|
Does the introduction provide sufficient background and include all relevant references? |
(x) |
( ) |
( ) |
( ) |
|
Is the research design appropriate? |
( ) |
(x) |
( ) |
( ) |
|
Are the methods adequately described? |
(x) |
( ) |
( ) |
( ) |
|
Are the results clearly presented? |
( ) |
(x) |
( ) |
( ) |
|
Are the conclusions supported by the results? |
( ) |
(x) |
( ) |
( ) |
Comments and Suggestions for Authors
Reviewer Comments / Remarks
General Comments
===============
The submitted paper deals with the problem of emotion recognition from speech using a fusion of multi- language corpora. In particular the authors propose the use of ensemble learning based on an algorithm that uses Random decision forest, Adaboost, LOgistic regression, and Gradient boosting machine (RALOG) presented in authors' previous works. Speech is analyzed to extract a list of spectral and procodic descriptors, and the RF-RFE algorithm was used to select the most important ones.
Experiments have shown that the proposed system can be used to succesfully classify negative / positive valence relatively quickly and accurately compared to other state of the art methods using a database consisting of five speech databases and speakers with four different languages.
Overall the quality of the paper is good, well-structured, with good use of English. The authors have key-pointed their contribution as well as the novelty of their proposed research. Experiments are extensive and the results have been compared to a number of well-known valence recognition methods from the literature.
Specific Comments to the authors
=========================
1. Line 350. The authors should include the percentage splitting the have used for the training / validation / testing procedure in each one of the 10-fold cross validation procedure.
Response:
We have used 80% of experimental utterances for training and 20% for testing and added a statement (line 371) to highlight the ratio used.
Lines 352-359. The authors should not relate the size of the stored features (in MB) after feature extraction with the "curse of dimensionality". The curse of dimensionality has to do with the number of the different descriptors that are used in the pattern recognition problem and not the demands for their storage on the disk.
Response:
We have removed Table 4 describing the change in size of the stored features and reference to the curse of dimensionality.
Line 372. What exactly does the processing time in msec that the authors have calculated and included in table 5 refer to? They should give more detail on what does (line 364) "time to process the data" mean.
Response:
The processing time refers to the total amount of time it took to train the algorithms before and after feature scaling using Rf-RFE. We have modified line 373-374 and changed processing time to training time.
- Line 437. Have the authors used only 100 speech utterances to test their algorithm from 10094 utterances (line 229)? If so, they should test the performance of their algorithm using a much larger subset of the sample files with positive/negative valence values (~10% -> 1000 utteraces). What is the distribution of the different languages in the test samples that were used? How many of them belong to the four different languages (German, Italian, British English, American etc)? The bigger test set that should be used, must include utterances from all different corpora (uniformly distributed).
Response:
No, we have used 2019 utterances that translated to 20% of the total utterances (10094). The explanation of the distribution of different languages have been include in Table 2 with explanation.
- How do the paper's results and the efficacy of the proposed system compare to each different corpora on its own? Are the recognition results similar for each corpora and the four different speakers? Are any differences in the recognition accuracy in specific languages observed? The authors should definately investigate this and include their findings in the paper.
Response:
The results of the performance of our proposed algorithm on individual corpus have been added to the manuscript (lines 478-511).
Reviewer 2 Report
The abstract is quite long. Only a brief summary of the most important details must be included.
The idea of considering corpora of different languages is interesting, however more details for supporting such an approach are missing. For example, why did the authors use languages coming from different language roots at the same time?
Why did the authors decide to include "call center" as a keyword?
The dimensional model of emotions considers more dimentions than "arousal", "dominance", and "valence".
How can be measured the impact of having recordings performed by actors in simulated scenarios?
Are the instances balanced in terms of gender?
Considering the Workflow in Figure 2, What do the authors mean by "Finish?"?
Did the authors evaluate the results using Information Gain?
The plot in Figure 4 must be improved.
In the Figure 7, what do the authors mean by "new dataset"?
It is not needed to describe well-known concepts such as "Stacking", "Accuracy", "Precision", "Recall", etc.
How is evaluated the "Promising feature?" in Figure 8?
What is "machine intelligent"? (Line 355)
Why did the authors consider that is important to describe the size in MB of the corpora in a Table?
Why is relevant to include the processing time in ms in a Table?
Tables 6, 7, and 8 need to be presented in a different format in order to improve the readability of the paper.
Regarding the References, I suggest to review whether or not is needed to have all of them, there are 5 pages of this section, that are too many for a research paper where an experimental setting was applied over a state-off-the-art corpora.
Author Response
Reviewer 2
Open Review
(x) I would not like to sign my review report
( ) I would like to sign my review report
English language and style
( ) Extensive editing of English language and style required
( ) Moderate English changes required
(x) English language and style are fine/minor spell check required
( ) I don't feel qualified to judge about the English language and style
|
Yes |
Can be improved |
Must be improved |
Not applicable |
|
|
Does the introduction provide sufficient background and include all relevant references? |
( ) |
(x) |
( ) |
( ) |
|
Is the research design appropriate? |
( ) |
(x) |
( ) |
( ) |
|
Are the methods adequately described? |
( ) |
( ) |
(x) |
( ) |
|
Are the results clearly presented? |
( ) |
( ) |
(x) |
( ) |
|
Are the conclusions supported by the results? |
( ) |
(x) |
( ) |
( ) |
Comments and Suggestions for Authors
The abstract is quite long. Only a brief summary of the most important details must be included.
Response:
We reduced the number of words in the Abstract to 247 words.
The idea of considering corpora of different languages is interesting, however more details for supporting such an approach are missing. For example, why did the authors use languages coming from different language roots at the same time?
Response:
We have included more practical statements (lines 80-85) to strengthen the idea of cross-language emotion recognition.
Why did the authors decide to include "call center" as a keyword?
Response:
We have replaced call center with ensemble learning as a new keyword.
The dimensional model of emotions considers more dimensions than "arousal", "dominance", and "valence".
Response:
We have changed our statement regarding the dimensional model to read “The dimensional approach presents speech emotions in different dimensions such as three dimension of arousal, dominance and valence [1,2]”.
How can be measured the impact of having recordings performed by actors in simulated scenarios?
Response:
Sometimes it is difficult to simulate the exact emotion verbally. However, we have used acted corpora because it was freely available for research works. Non-acted corpora are difficult to access because of a variety of reasons such as the rigorous process of ethics that a researcher must go through, which can be frustrating. In addition, the experimental corpora have made it easy for us to benchmark our methods with other works done in the past.
Are the instances balanced in terms of gender?
Response:
The instances are indeed imbalanced in terms of gender, because the SAVEE corpus contains only male speech and CREMA-D contains more male speech than female speech. The three other corpora are gender balanced. We have included the details in the paper (lines 231-240).
Considering the Workflow in Figure 2, What do the authors mean by "Finish?"?
Response:
The word Finish has been corrected in Figure 2 to convey the real message of the flow. It now reads “Is the recording of prediction errors at each level finished?”
Did the authors evaluate the results using Information Gain?
Response:
We did not use Information Gain. Instead, we used recursive feature elimination riding on random forest.
The plot in Figure 4 must be improved.
Response:
We have inserted a bigger and more visible plot in Figure 4.
In the Figure 7, what do the authors mean by "new dataset"?
Response:
With regards to Figure 7, “new dataset” refers to the testing data.
It is not needed to describe well-known concepts such as "Stacking", "Accuracy", "Precision", "Recall", etc.
Response:
We have removed the descriptions of "Stacking", "Accuracy", "Precision", "Recall".
How is evaluated the "Promising feature?" in Figure 8?
Response:
The promising features in Figure 8 are evaluated using the scores shown in the graph in Figure 4.
What is "machine intelligent"? (Line 355)
Response:
We have replaced machine intelligent with machine learning. It was a typo because the hands are not perfect.
Why did the authors consider that is important to describe the size in MB of the corpora in a Table?
Response:
We have removed all the issues to do with the size of the data.
Why is relevant to include the processing time in ms in a Table?
Response:
We have used processing time (now changed to training time) as a performance measure in this research because we believe it is important to consider speed when evaluating the performance of algorithms. Moreover, we have considered it because it has been successfully used by other researchers in other papers that we have referenced.
Tables 6, 7, and 8 need to be presented in a different format in order to improve the readability of the paper.
Response:
We have designed Tables 6,7 and 8 in that fashion because it was difficult to do it any other way. One alternative was to split the Tables, but that would have resulted in a huge number of tables making readability even more difficult. Hence, we have followed the format in our previous paper in algorithms.
Regarding the References, I suggest to review whether or not is needed to have all of them, there are 5 pages of this section, that are too many for a research paper where an experimental setting was applied over a state-off-the-art corpora.
Response:
We have removed a few references which had to do with the descriptions of "Stacking", "Accuracy", "Precision", "Recall". We could not remove the other references because they are important for this paper even though it was an experimental setting. We had to honestly support our ideas with relevant concepts by other researchers to build strong cases. We have considered references a utility for our work.
Round 2
Reviewer 1 Report
The authors have included all the revewer's comments in their manuscript and corrected any parts that needed correction.
Author Response
Reviewer 1
(x) I would not like to sign my review report
( ) I would like to sign my review report
English language and style
( ) Extensive editing of English language and style required
( ) Moderate English changes required
(x) English language and style are fine/minor spell check required
( ) I don't feel qualified to judge about the English language and style
|
Yes |
Can be improved |
Must be improved |
Not applicable |
|
|
Does the introduction provide sufficient background and include all relevant references? |
(x) |
( ) |
( ) |
( ) |
|
Is the research design appropriate? |
(x) |
( ) |
( ) |
( ) |
|
Are the methods adequately described? |
(x) |
( ) |
( ) |
( ) |
|
Are the results clearly presented? |
(x) |
( ) |
( ) |
( ) |
|
Are the conclusions supported by the results? |
(x) |
( ) |
( ) |
( ) |
Comments and Suggestions for Authors
The authors have included all the revewer's comments in their manuscript and corrected any parts that needed correction.
Response
We sincerely appreciate the reviewer for the positive comments.
Reviewer 2 Report
The paper has been improved from the last version.
The information in Figure 4 NEEDS to be presented in a different format. Such a big plot is not appropriate for a research paper.
The "promising feature" sounds inadequate for a research paper, modify such words.
Tables 6-9 can be actually presented in a different (and more readable) format, the authors need to figure out a better distribution of their results.
I still consider that there are TOO MANY references, for example, when mentioning the "categorical model of emotions", the authors cited two references, however, any of them are about this topic, like for example the Plutchik model of the one described in Reference 15.
Avoid answering a reviewer in an impolite way: "hands are not perfect" ...
Author Response
Reviewer 2
Open Review
(x) I would not like to sign my review report
( ) I would like to sign my review report
English language and style
( ) Extensive editing of English language and style required
( ) Moderate English changes required
(x) English language and style are fine/minor spell check required
( ) I don't feel qualified to judge about the English language and style
|
Yes |
Can be improved |
Must be improved |
Not applicable |
|
|
Does the introduction provide sufficient background and include all relevant references? |
( ) |
( ) |
(x) |
( ) |
|
Is the research design appropriate? |
(x) |
( ) |
( ) |
( ) |
|
Are the methods adequately described? |
(x) |
( ) |
( ) |
( ) |
|
Are the results clearly presented? |
( ) |
( ) |
(x) |
( ) |
|
Are the conclusions supported by the results? |
( ) |
( ) |
(x) |
( ) |
Comments and Suggestions for Authors
The paper has been improved from the last version.
Response 1:
We sincerely appreciate the reviewer for the positive comments.
The information in Figure 4 NEEDS to be presented in a different format. Such a big plot is not appropriate for a research paper.
Response 2:
We have changed figure 4 to make it more visible and more appropriate for a research paper.
The "promising feature" sounds inadequate for a research paper, modify such words.
Response 3:
We have replaced the word “promising” with discriminative. Furthermore, we have reduced the abstract from 247 words to 210 words.
Tables 6-9 can be actually presented in a different (and more readable) format, the authors need to figure out a better distribution of their results.
Response 4:
We have redesigned the tables to make them more readable and easier to comprehend.
I still consider that there are TOO MANY references, for example, when mentioning the "categorical model of emotions", the authors cited two references, however, any of them are about this topic, like for example the Plutchik model of the one described in Reference 15.
Response 5:
We have removed 33 references, so the paper now has 56 references only.
Avoid answering a reviewer in an impolite way: "hands are not perfect" ...
Response 6:
We sincerely apologize to the reviewer. We did not mean to answer the reviewer impolitely.

Round 3
Reviewer 2 Report
The paper has been improved from the last two versions.
I am sure that the research topic of this paper can be further explored.